# Multivariate Profiles of Female Sexual Function: A Cluster Analysis of FSFI Domains in Women with and Without PCOS

**DOI:** 10.3390/biomedicines13123069

**Published:** 2025-12-12

**Authors:** Andrei Daescu, Ana-Maria Cristina Daescu, Liana Dehelean, Dan-Bogdan Navolan, Alexandru-Ioan Gaitoane, Dana Stoian

**Affiliations:** 1Doctoral School Department, “Victor Babes” University of Medicine and Pharmacy, 300041 Timisoara, Romania; andrei.daescu@umft.ro; 2Neurosciences Department, Discipline of Psychiatry, “Victor Babes” University of Medicine and Pharmacy, 300041 Timisoara, Romania; 3Department of Obstetrics and Gynecology, “Victor Babes” University of Medicine and Pharmacy, 300041 Timisoara, Romania; navolan.dan@umft.ro; 4Department of Psychiatry, Timis County Emergency Clinical Hospital “Pius Brinzeu”, 300041 Timisoara, Romania; gaitoane.alexandru-ioan@hosptm.ro; 5Department of Internal Medicine II, Discipline of Endocrinology, “Victor Babes” University of Medicine and Pharmacy, 300041 Timisoara, Romania; stoian.dana@umft.ro

**Keywords:** PCOS, FSFI, female sexual function, BESAQ, body image, sexual dysfunction

## Abstract

**Background/Objectives:** Female sexual dysfunction (FSD) is common in PCOS, yet mean group comparisons can mask variability at the individual level. We aimed to identify and characterize person-centered profiles of sexual function from the six FSFI (Female Sexual Function Index) domains, and secondarily to describe the distribution of PCOS across profiles. **Methods**: In an age- and anthropometry-matched case–control sample, unsupervised clustering on FSFI domains was performed; clinical and psychosocial correlates were compared, and logistic regression tested prediction of FSFI-defined FSD. **Results**: Two profiles emerged—Sexual Dysfunction vs. Preserved Function—with clear multivariate separation. Dysfunction showed lower FSFI, higher adiposity, and worse body-image discomfort. PCOS was more frequent in Dysfunction but not significantly. Cluster membership predicted FSD. **Conclusions**: Person-centered profiling reveals clinically meaningful heterogeneity that transcends diagnosis and highlights adiposity and body-image distress as salient, potentially modifiable correlates.

## 1. Introduction

Polycystic ovary syndrome (PCOS) is a prevalent and multifaceted endocrine disorder, affecting between 5% and 20% of women of reproductive age, contingent on the diagnostic criteria applied [1]. Defined by the presence of at least two out of the following three criteria—hyperandrogenism, ovulatory dysfunction, and polycystic ovarian morphology, PCOS manifests in a heterogeneous clinical spectrum that encompasses reproductive, metabolic, dermatological, and psychological domains [2,3]. Beyond its physiological consequences, PCOS profoundly disrupts women’s emotional well-being, self-image, and quality of life, prompting an urgent need for holistic, evidence-based approaches to its evaluation and management [4].

An increasingly recognized yet insufficiently addressed component of PCOS is female sexual dysfunction (FSD). Mounting evidence indicates that women with PCOS are more likely to experience impairments across multiple facets of sexual function—including desire, arousal, lubrication, orgasm, satisfaction, and pain—compared to women without the syndrome [5,6,7]. These impairments are not only prevalent but also clinically significant, impacting intimate relationships and mental health [8]. Studies indicate that phenotype-specific differences in clinical and biochemical features of PCOS may modulate the severity and presentation of FSD, underscoring the importance of nuanced phenotypic analysis [9].

The role of androgens in female sexual function is particularly complex in PCOS. While testosterone is known to play a facilitative role in sexual desire, both hyperandrogenism and hypoandrogenism have been implicated in sexual dysfunction among PCOS patients, suggesting a U-shaped association [10]. Furthermore, metabolic complications such as insulin resistance, obesity, and dyslipidemia—common in PCOS—contribute to hormonal imbalance, reduced vascular perfusion, and heightened systemic inflammation, all of which negatively influence sexual health [11].

Another critical yet often underappreciated factor contributing to FSD in PCOS is body image dissatisfaction. Clinical symptoms such as hirsutism, acne, androgenic alopecia, and central adiposity can significantly impair self-perception, especially in intimate contexts [12]. Negative body image fosters avoidance behaviors, inhibits sexual assertiveness, and increases anxiety during sexual activity [13]. Instruments such as the Body Exposure during Sexual Activities Questionnaire (BESAQ) have been validated to measure these concerns and have demonstrated strong inverse correlations with scores on the Female Sexual Function Index (FSFI), highlighting the role of body uneasiness in sexual avoidance and dissatisfaction [14,15,16,17].

Psychological comorbidities further amplify the burden of FSD in PCOS. A substantial proportion of women with PCOS report symptoms of depression, anxiety, and low self-esteem—conditions that independently predict poorer sexual function [12,18,19]. These psychological disturbances may stem from both the chronic nature of PCOS and the distressing physical alterations it causes. Additionally, the challenges of infertility and social stigma related to physical appearance contribute to emotional strain and impaired sexual intimacy [20].

In a prior study involving 54 women diagnosed with PCOS, we explored the interrelations between sexual function and a range of hormonal, anthropometric, and psychometric parameters [21]. That investigation revealed that BMI, free testosterone, the LH/FSH ratio, and BESAQ scores were all independent predictors of FSD. Notably, we identified a BESAQ score threshold of 1.97 as a clinically relevant cut-off for distinguishing women with sexual dysfunction. Despite these insights, the absence of a control group in that study restricted our ability to compare findings against a normative baseline, limiting generalizability.

Against this backdrop, person-centered approaches that uncover latent patterns of sexual functioning can overcome the limitations of relying solely on the FSFI total score and simple case–control contrasts. Building on our prior observations—where body image (BESAQ), body mass, and hormonal markers were independently associated with sexual dysfunction but in the absence of a matched control group—we designed an age- and anthropometry-matched study to map profiles of female sexual function across the six FSFI domains. Our primary aim was to identify and validate distinct profiles of female sexual function using cluster analysis of FSFI domains; our secondary aim was to describe how PCOS status is distributed across these profiles and to examine key clinical and psychosocial correlates (BMI, waist circumference, BESAQ) with implications for risk assessment and clinical stratification.

We employed cluster analysis because it enables the identification of latent profiles of sexual function beyond mean differences, capturing heterogeneity that traditional group comparisons cannot detect.

## 2. Materials and Methods

### 2.1. Study Design and Population

The study included 68 women, of whom 34 had a clinical diagnosis of polycystic ovary syndrome (PCOS) and 34 were controls. Propensity score matching was used to minimize baseline differences in age, body mass index (BMI), and abdominal circumference (AC), ensuring comparability between groups. Matching was performed using a nearest-neighbor algorithm with a caliper constraint, which may allow minor residual imbalances but improves overall covariate balance.

All PCOS participants met at least two of the Rotterdam criteria: clinical or biochemical hyperandrogenism, ovulatory dysfunction, and polycystic ovarian morphology. The control group consisted of women of reproductive age (18 to 40 years old), without PCOS or other endocrine disorders, matched by age and body mass index (BMI) to the PCOS group. Inclusion criteria for controls included regular menstrual cycles (21–35 days), absence of hirsutism or acne, and no history of infertility or gynecological endocrinopathies. Exclusion criteria for both groups included pregnancy, known psychiatric disorders, thyroid dysfunction, previously established diagnosis of diabetes mellitus, or use of medication that could interfere with the results. In this study, participants were not subclassified into Rotterdam phenotypes (A–D) due to incomplete availability of concurrent hormonal and ultrasound features needed to define all subtypes. Diagnostic status was therefore used only to describe its distribution across sexual function profiles, rather than to model phenotype-specific effects.

The primary aim of the study was to identify and characterize distinct subgroups of women based on their sexual function profiles, derived from the six domains of the Female Sexual Function Index (FSFI: desire, arousal, lubrication, orgasm, satisfaction, and pain). Diagnostic status (PCOS vs. control) was considered only secondarily, to describe the distribution of participants across the identified profiles, rather than to compare groups directly.

### 2.2. Data Collection and Instruments

Demographic and clinical data were collected from both groups, including age, living environment (urban/rural), educational level, marital and occupational status, religion, BMI, and abdominal circumference.

Sexual function was assessed using the Female Sexual Function Index (FSFI), a 19-item validated instrument that evaluates six domains: desire, arousal, lubrication, orgasm, satisfaction, and pain. The Romanian version of FSFI was used in both groups, with a total score below 26.55 indicating sexual dysfunction. The FSFI has demonstrated excellent internal consistency (Cronbach’s alpha > 0.9 across domains) and strong test–retest reliability. The Romanian version used in this study has also shown good construct validity and psychometric performance in local samples [16,17,22]. In line with FSFI scoring requirements, participants were eligible only if they had been sexually active within the past 4 weeks; thus, relationship status (married, cohabiting, dating, or single) did not influence inclusion.

Body image perception in the sexual context was evaluated using the Body Exposure during Sexual Activities Questionnaire (BESAQ). We used the short form of the BESAQ that consists of 18 items rated on a Likert-type scale (from 0 to 4), yielding a total score by summing all item responses, with some reverse-scored items. Total score reflects the degree of anxiety and avoidance behaviors related to one’s body during intimacy. Higher scores denote increased discomfort. The BESAQ has shown high internal consistency (Cronbach’s alpha = 0.88) and good convergent validity, particularly with measures of sexual avoidance, anxiety, and body image concerns [15].

The present analysis did not include laboratory indices of androgen status (e.g., total/free testosterone, SHBG), insulin resistance (e.g., fasting insulin, HOMA-IR), or lipid profiles. Anthropometric markers (BMI, abdominal circumference) were used as pragmatic proxies for metabolic burden given their clinical availability and strong empirical links with sexual function.

### 2.3. Ethical Considerations

The study was performed in accordance with the Ethical Guidelines of the Helsinki Declaration and was approved by the Ethics Committee of Victor Babes University of Medicine and Pharmacy, Timisoara, Romania. All subjects agreed to the evaluation and provided their written informed consent prior to inclusion.

### 2.4. Statistical Analysis

All analyses were performed using R, version 4.3.0 (R Core Team, R Foundation for Statistical Computing, Vienna, Austria; 2024). Data processing and coding were conducted in RStudio, version 2023.06.0+421 (Posit Software, PBC, Boston, MA, USA; 2023), an integrated development environment for R. Statistical significance was set at *p* < 0.05, two-tailed.

K-means clustering was conducted on FSFI domain scores to derive natural groupings of sexual function. We selected *k*-means clustering due to its simplicity, interpretability, and suitability for continuous, multivariate data such as FSFI domain scores. Compared to hierarchical clustering, *k*-means is computationally efficient and well-suited for identifying spherical, compact clusters in moderate sample sizes [23]. Latent profile analysis (LPA) and Gaussian mixture models were considered but were not implemented due to the limited sample size and the risk of model overfitting. Internal validation (silhouette scores, bootstrapped Jaccard indices, and MANOVA/PERMANOVA) confirmed the stability and distinctiveness of the *k* = 2 solution. The optimal number of clusters was determined using both the elbow method, which evaluates reductions in within-cluster sum of squares, and the gap statistic, which compares observed clustering with reference datasets. To assess internal validity, silhouette analysis was used to quantify cohesion within clusters and separation between them. Stability of the clustering solution was further tested with bootstrap resampling, calculating the Jaccard index for each cluster, with values above 0.70 considered acceptable reproducibility. Cluster stability was evaluated using 1000 bootstrap resamples, and Jaccard indices > 0.70 were considered indicative of acceptable reproducibility.

To test whether the clusters represented distinct multivariate profiles, we conducted a multivariate analysis of variance (MANOVA) across FSFI domains, summarized with Pillai’s trace. A non-parametric permutational MANOVA (PERMANOVA) was also performed to confirm robustness of the clustering solution without distributional assumptions.

Cluster visualization was achieved with principal component analysis (PCA), projecting FSFI domain scores into two dimensions. Cluster centroids and 68% confidence ellipses were added to illustrate group separation. We used 68% confidence ellipses, which approximate ±1 standard deviations in multivariate normal space, to visually depict the core region of each cluster and illustrate centroid separation.

To characterize clinical and psychosocial differences between clusters, we compared BMI, abdominal circumference, menstrual cycle interval, and body image scores (measured by the Body Exposure in Sexual Activities Questionnaire, BESAQ). Continuous variables were compared using independent-sample *t*-tests, while categorical variables were assessed with Chi-squared tests.

The clinical relevance of the identified profiles was examined with logistic regression, testing whether cluster membership predicted sexual dysfunction, defined as FSFI ≤ 26.55. Model performance was evaluated using Nagelkerke’s R^2^.

Finally, Spearman’s rank correlation was used to examine associations between FSFI scores, body image (BESAQ), anthropometric parameters (BMI and AC), and age, to contextualize the mechanisms underlying cluster separation.

A post hoc power analysis was performed to evaluate the statistical sensitivity of the sample to detect the observed effects. For the primary comparison of FSFI total scores between the two cluster-derived profiles, we computed Cohen’s *d* based on the observed means and standard deviations. The analysis was conducted using a two-tailed independent-samples *t*-test with an alpha level of 0.05.

For the logistic regression model assessing the association between cluster membership and sexual dysfunction (FSFI ≤ 26.55), the effect size was estimated using Nagelkerke’s R^2^ and converted to Cohen’s *f*^2^ for compatibility with power analysis procedures.

## 3. Results

### 3.1. Descriptive Characteristics of the Study Population

The final analytic sample consisted of 68 women, divided equally between the PCOS group (*n* = 34) and the control group (*n* = 34) following propensity score matching for age, body mass index (BMI), and abdominal circumference (AC). This matching procedure was applied to minimize baseline anthropometric differences between groups and to ensure comparability with respect to key clinical covariates.

The median age of participants was 24 years (interquartile range [IQR]: 23–28), and the median BMI was 24.6 kg/m^2^ (IQR: 22.0–27.0). Slightly more than half of the women (51.5%) met the clinical threshold for female sexual dysfunction (FSD), operationalized as a total FSFI score ≤26.55, indicating that sexual dysfunction was common even in this relatively young and educated cohort.

Sociodemographic characteristics showed that most participants resided in urban environments (75%) and had completed higher education (72%). Employment status was nearly evenly split, with 53% of participants employed at the time of assessment. The majority of women were unmarried (62%), reflecting the young age distribution of the sample. Clinical and psychosocial characteristics, including menstrual cycle interval (MCI) and body image scores (BESAQ), are detailed in Table 1, Table 2 and Table 3.

### 3.2. Cluster Analysis of Sexual Function Profiles

#### 3.2.1. Determination of Optimal Clusters

To explore heterogeneity in sexual function, we performed an unsupervised k-means cluster analysis based on the six FSFI domains (desire, arousal, lubrication, orgasm, satisfaction, and pain).

The elbow method suggested that a two-cluster solution was most appropriate, as the curve displayed a clear inflection at *k* = 2, after which further partitions yielded only minimal improvement in model fit (Figure 1).

The gap statistic formally identified *k* = 1 as the most conservative solution, reflecting the modest sample size and some overlap in sexual function scores. However, closer inspection of the gap curve revealed a dip at *k* = 2 followed by a steady upward trend across higher cluster numbers (Figure 2). This pattern is consistent with the presence of a meaningful two-group structure, even if the automated rule favored a single group.

Taken together, the elbow method and the shape of the gap curve suggested that *k* = 2 provided a parsimonious and clinically interpretable solution. This conclusion was reinforced by subsequent validation analyses (Section 3.2.2), which confirmed that the two clusters were both stable and clinically meaningful.

#### 3.2.2. Validation and Stability of the Two-Cluster Solution

The robustness of the two-cluster solution was examined using several complementary approaches.

First, silhouette analysis (Figure 3) indicated moderate separation overall (mean coefficient = 0.31). The first cluster showed lower internal cohesion (average silhouette = 0.14), whereas the second cluster was more compact and internally consistent (0.38). This pattern suggests that one group was more heterogeneous in its sexual function scores, while the other was more uniform.

Second, bootstrapped resampling (Table 4) demonstrated that the two clusters were reproducible. The Jaccard stability index was 0.90 for one cluster (very high stability) and 0.75 for the other (acceptable stability). By convention, values above 0.70 indicate that the clusters are not statistical artifacts but represent meaningful groupings.

Finally, multivariate testing (Table 5) confirmed that the clusters were statistically distinct. MANOVA showed a strong overall effect of cluster membership across the six FSFI domains (Pillai’s trace = 0.69, F (6, 61) = 22.6, *p* < 0.001). This finding was corroborated by a non-parametric PERMANOVA, which indicated that nearly one-third of the variance in sexual function scores (R^2^ = 0.29) was explained by cluster membership (F (1, 66) = 26.7, *p* < 0.001).

Together, these validation methods support the reliability of the two-cluster solution, showing that the subgroups are both stable and clinically relevant, even though they differ in their degree of internal cohesion.

#### 3.2.3. Visualization of Cluster Separation

To further examine the structure of the two-cluster solution, we performed a principal component analysis (PCA) on the six FSFI domains. PCA reduced the multidimensional data into two principal components that together captured the majority of the variance in sexual function profiles.

As illustrated in Figure 4, the two clusters showed partial but clinically meaningful separation in this reduced space. Cluster 1 (red) encompassed women with greater dispersion around the centroid, reflecting heterogeneity in the severity and pattern of sexual dysfunction. In contrast, Cluster 2 (blue) was more compact and tightly grouped, indicating a relatively homogeneous profile of preserved sexual function. The 68% confidence ellipses and cluster centroids (black crosses) further highlighted the distinction between the two profiles.

For interpretability in subsequent sections, Cluster 1 will be referred to as the “Sexual Dysfunction Profile” and Cluster 2 as the “Preserved Function Profile”.

#### 3.2.4. Clinical and Psychosocial Profiles of Clusters

Comparisons between the two profiles revealed pronounced and statistically significant differences across both sexual and non-sexual parameters (Table 6).

Women in the Sexual Dysfunction Profile exhibited markedly lower total FSFI scores compared to those in the Preserved Function Profile (mean 17.9 ± 5.2 vs. 28.1 ± 3.0, *p* < 0.001), confirming that the primary distinction between clusters was widespread impairment across multiple sexual domains.

Participants in the Dysfunction Profile also had significantly higher BMI (29.1 ± 7.3 vs. 24.8 ± 4.5, *p* = 0.03) and greater abdominal circumference (93.9 ± 19.3 cm vs. 80.2 ± 11.7 cm, *p* = 0.009), suggesting an association between increased adiposity and reduced sexual function.

Body image dissatisfaction, as measured by the BESAQ, was nearly twice as high in the Dysfunction Profile compared to the Preserved Profile (2.48 ± 1.08 vs. 1.17 ± 0.85, *p* < 0.001), underscoring the psychosocial dimension of sexual health.

Women in the Dysfunction Profile also tended to have longer menstrual cycle intervals (99.1 ± 83.1 vs. 57.6 ± 55.5 days), although this difference narrowly missed statistical significance (*p* = 0.06).

No age difference was observed between profiles (27.3 vs. 25.6 years, *p* = 0.24).

Taken together, these findings delineate two clinically distinct subgroups: a Sexual Dysfunction Profile defined by impaired sexual function, higher adiposity, and poorer body image, and a Preserved Function Profile characterized by relatively intact sexual function and more favorable clinical and psychosocial characteristics.

#### 3.2.5. Sociodemographic Profiles of Clusters

Sociodemographic characteristics were broadly comparable between the Sexual Dysfunction Profile and the Preserved Function Profile (Table 7). Environment of residence, level of education, marital status, and employment status did not differ significantly between profiles. Although women from rural areas appeared more frequently in the Preserved Function Profile (88% vs. 12% in the Dysfunction Profile), this difference did not reach statistical significance (*p* = 0.11).

These findings indicate that cluster separation is not explained by sociodemographic background, but rather by clinical and psychosocial variables such as BMI, abdominal adiposity, and body image. In other words, the profiles identified reflect meaningful differences in health status and sexual function, rather than differences in social or educational circumstances.

#### 3.2.6. Predictive Value of Clusters for Sexual Dysfunction

To evaluate whether the identified profiles corresponded to clinically meaningful dysfunction, a logistic regression was performed using the established FSFI cut-off of 26.55 as the outcome criterion. Belonging to the Preserved Function Profile was associated with substantially lower odds of sexual dysfunction compared with the Sexual Dysfunction Profile (OR = −0.66, 95% CI −0.88 to −0.44, *p* < 0.001). The model demonstrated acceptable explanatory power, with a Nagelkerke R^2^ of 0.368 (Table 8).

These results confirm that the profiles derived from unsupervised clustering are not only statistically distinct but also clinically relevant, reliably distinguishing women at risk of sexual dysfunction from those with preserved sexual function.

### 3.3. Relationship Between Clusters and PCOS Status

Although the clustering procedure was based exclusively on FSFI domain scores, the distribution of diagnostic categories across profiles was not entirely random (Figure 5). Women with PCOS were more frequently classified into the Sexual Dysfunction Profile (61% vs. 39%), whereas controls were somewhat more likely to belong to the Preserved Function Profile (54% vs. 46%). This difference, however, did not reach statistical significance (χ^2^ = 1.21, *p* = 0.27).

These results indicate that while PCOS status may increase the likelihood of being classified into the dysfunction profile, it is not the sole determinant of profile membership. Clinical and psychosocial characteristics—particularly BMI, abdominal adiposity, and body image—appear to exert an equally important, if not greater, influence. This underscores the added value of the clustering approach, which captures heterogeneity in sexual function that extends beyond diagnostic boundaries.

### 3.4. Correlates of Sexual Function and Cluster Membership

Spearman correlation analyses provided additional evidence for the influence of body composition and psychosocial factors on sexual function (Table 9).

Higher BMI was inversely correlated with overall sexual function, as reflected by total FSFI scores (ρ = −0.374, *p* = 0.002), and particularly with the arousal domain (ρ = −0.474, *p* < 0.001).

Menstrual cycle interval (MCI) was also negatively associated with arousal (ρ = −0.365, *p* = 0.002).

Abdominal circumference also demonstrated a significant negative association with arousal (ρ = −0.362, *p* = 0.002).

In terms of psychosocial factors, body image dissatisfaction, measured by the BESAQ, was strongly and consistently linked with poorer sexual function, showing robust negative correlations with total FSFI scores (ρ = −0.580, *p* < 0.001) and nearly all subdomains, with the strongest effect observed for arousal (ρ = −0.645, *p* < 0.001).

Age was also associated with sexual function, as sexual desire declined modestly with increasing age within this relatively young cohort (ρ = −0.357, *p* = 0.003).

Taken together, these correlations reinforce the cluster-based findings: women classified into the Sexual Dysfunction Profile were more likely to exhibit higher adiposity and greater body image concerns, both of which were consistently linked with poorer sexual function. This suggests that psychosocial and anthropometric factors act as key determinants of profile membership, complementing diagnostic status.

### 3.5. Post Hoc Power Analysis

To evaluate the adequacy of the sample size for the primary comparisons, a post hoc power analysis was conducted based on observed effect sizes from the study (Table 10). The comparison of FSFI total scores between the two identified profiles (Sexual Dysfunction vs. Preserved Function) revealed a large effect size (Cohen’s d = 2.74), based on group means and standard deviations reported in Table 6. This yielded a power of 1.00 (100%) for detecting a significant difference at α = 0.05.

For the logistic regression model predicting female sexual dysfunction (FSFI ≤ 26.55) based on cluster membership, the effect size was estimated using the reported Nagelkerke R^2^ = 0.368. This corresponds to a Cohen’s f^2^ of 0.582 and a standardized effect size of 0.763. Using these values, the estimated statistical power was 0.878 (87.8%), confirming adequate sensitivity of the sample to detect this effect.

## 4. Discussion

This study extends person-centered analyses of the FSFI by identifying two reproducible profiles of sexual function—one characterized by widespread impairment and one by preserved function—in women with and without PCOS. Framing sexual function as a multivariate construct moves beyond case–control averages and emphasizes heterogeneity that cuts across diagnostic boundaries.

Prior work has typically treated PCOS as a homogeneous group, comparing mean scores against controls. Our clustering shows substantial heterogeneity even within PCOS: while women with PCOS were more often in the Dysfunction profile, diagnostic status did not significantly determine profile membership once domain patterns were considered. Across profiles, higher adiposity and greater body-image discomfort aligned with poorer sexual function, consistent with evidence that metabolic burden and appearance-related distress shape desire, arousal, lubrication, satisfaction, and pain. Notably, abdominal circumference appeared to differentiate the clusters more clearly than BMI, suggesting that central adiposity—more closely linked to visceral fat and cardiometabolic risk—may play a more direct role in sexual function than overall body mass. This distinction aligns with evidence that visceral fat exerts stronger effects on vascular, hormonal, and inflammatory pathways that may impair arousal and lubrication.

Mechanistically, several pathways may explain the link between adiposity, body-image discomfort, and impaired arousal and lubrication. Central adiposity is associated with endothelial dysfunction, reduced genital blood flow, and low-grade systemic inflammation, all of which are known to negatively affect sexual arousal and vaginal lubrication. Additionally, adiposity contributes to altered estrogen-androgen balance and reduced SHBG, which can further impair sexual responsiveness. Beyond biological mechanisms, body-image distress has been shown to increase self-monitoring, avoidance behavior, and performance anxiety during intimacy, which selectively disrupt the arousal–lubrication phase of the sexual response cycle. These converging vascular, hormonal, and psychosocial mechanisms likely contribute to the cluster differentiation observed in our sample [24,25].

Conceptually, these functional profiles parallel unsupervised PCOS subtyping based on endocrine/metabolic traits, but they are derived solely from sexual-function domains. Any apparent concordance should be regarded as hypothesis-generating rather than evidence of shared biological pathways [26].

While the overall sample size was modest (*n* = 68), it was sufficient for the primary analyses. A post hoc power analysis indicated that the study had 100% power to detect the large difference in FSFI total scores between the identified profiles (Cohen’s d = 2.74), and 87.8% power to detect the observed effect in the logistic regression model (Cohen’s f^2^ = 0.582, Nagelkerke R^2^ = 0.368). These results provide confidence in the validity of the key findings. Nonetheless, small samples may attenuate the stability of unsupervised clustering techniques, limit the detection of subtle subgroup effects, and reduce generalizability. Future studies should replicate these findings in larger, more diverse populations and include biological and longitudinal markers to further refine clinical phenotyping.

An important limitation of this study is the modest sample size (*n* = 68), which may reduce the stability and generalizability of clustering solutions. Although internal validation indices (silhouette coefficients, bootstrapped Jaccard stability, MANOVA/PERMANOVA) supported the robustness of the two-cluster structure, replication in larger and more diverse samples is warranted. Furthermore, the study population was relatively young, highly educated, and predominantly urban, which may limit the external validity of the findings in older, less educated, or rural populations. Therefore, the identified profiles should be interpreted as preliminary and require confirmation in broader demographic and cultural contexts. Given the cross-sectional design, the associations between adiposity, body-image discomfort, and sexual function should be interpreted as correlational; the temporal or causal direction of these relationships cannot be determined. In addition, the possibility of unmeasured confounding cannot be excluded; factors such as relationship quality, depressive or anxiety symptoms, and other psychological comorbidities were not assessed and may contribute to variations in sexual functioning independently of adiposity or body-image concerns. Moreover, the absence of hormonal (e.g., testosterone, SHBG) and metabolic markers (e.g., fasting glucose, insulin, HOMA-IR) limits mechanistic interpretation; the profiles identified here reflect functional patterns rather than underlying endocrine or metabolic phenotypes. Future studies incorporating endocrine and insulin-resistance indices will be essential to clarify potential biological pathways.

A methodological limitation concerns the determination of the optimal number of clusters. While the elbow method and clinical interpretability supported a two-cluster solution, the gap statistic indicated k = 1 as the most conservative choice. This discrepancy likely reflects the modest sample size and partial overlap in FSFI domain scores, which tends to bias the gap statistic toward parsimony. Although internal validation indices (Jaccard stability, MANOVA/PERMANOVA) provided support for the two-cluster structure, the lower silhouette coefficient observed for the Sexual Dysfunction profile indicates greater within-cluster heterogeneity. This suggests that the dysfunction group may encompass multiple subtypes that were not detectable with the present sample size. Future studies with larger and more diverse samples should explore whether more granular subgroups of sexual dysfunction can be reliably identified.

From a clinical perspective, the two profiles identified here may offer a preliminary framework for individualized care. Women in the Dysfunction profile who exhibit high body-image discomfort may benefit from psychosexual or cognitive-behavioral interventions targeting avoidance, appearance-related distress, and self-focus during sexual activity. Conversely, women whose profile is characterized by higher adiposity and prominent arousal/lubrication difficulties may respond better to interventions aimed at metabolic optimization, lifestyle modification, or hormonal evaluation. Although these applications remain exploratory, person-centered profiling could help clinicians tailor counseling and treatment priorities more effectively.

External validation in larger, multicenter cohorts should test generalizability, stability over time, and transportability across settings. Adding androgen profiles and insulin-resistance markers will probe biological pathways linking adiposity and arousal/lubrication deficits. Finally, prospective studies should evaluate whether profile-tailored interventions (psychosexual strategies for high-BESAQ phenotypes; metabolic optimization for high-adiposity phenotypes) can shift women toward a Preserved profile and improve patient-centered outcomes.

## 5. Conclusions

Using a person-centered, unsupervised clustering of FSFI domains in an age- and anthropometry-matched sample of women with and without PCOS, we identified two robust profiles—a Sexual Dysfunction profile and a Preserved Function profile—that cut across diagnostic boundaries. While women with PCOS were more often represented in the Dysfunction profile, diagnosis alone did not determine membership; instead, higher adiposity and greater body-image discomfort emerged as salient correlates of poorer sexual function. These findings challenge homogeneous views of PCOS and support routine, integrated assessment of FSFI domains alongside anthropometric and body-image measures. Overall, profile-informed screening and targeted counseling may better align care with patients’ needs than diagnosis-based approaches alone.

## Figures and Tables

**Figure 1 biomedicines-13-03069-f001:**
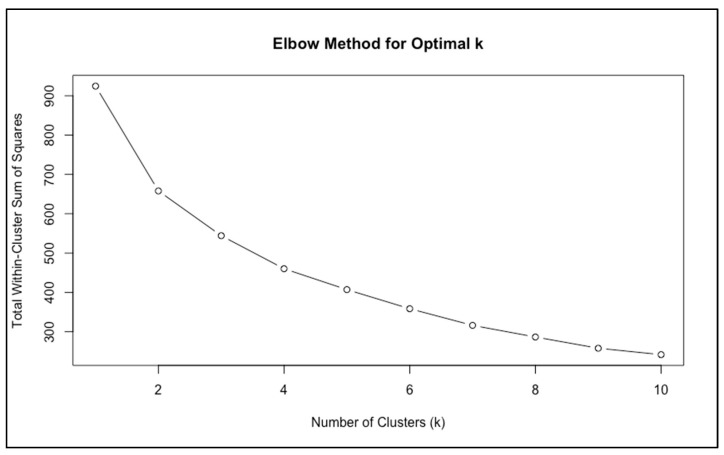
Elbow Plot Indicating Optimal Number of Clusters.

**Figure 2 biomedicines-13-03069-f002:**
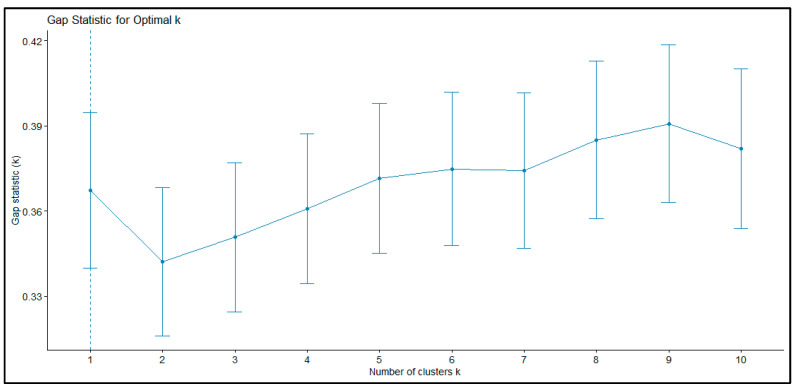
Gap Statistic Curve for Determining the Optimal Number of Clusters.

**Figure 3 biomedicines-13-03069-f003:**
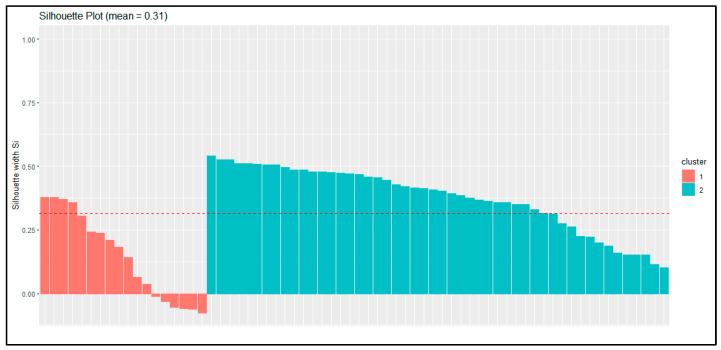
Silhouette Analysis of Cluster Cohesion and Separation. The red dotted line represents average silhouette score across all clusters.

**Figure 4 biomedicines-13-03069-f004:**
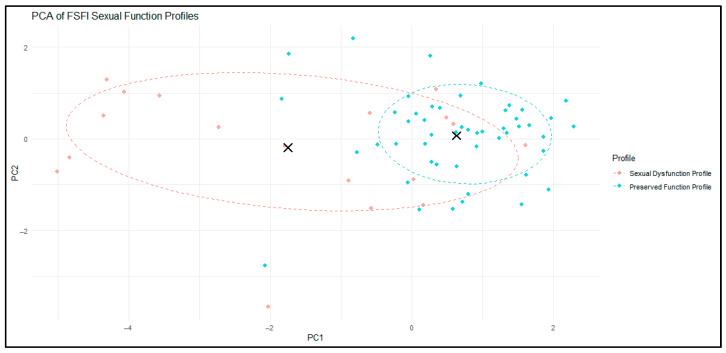
PCA-Based Visualization of Clusters Derived from FSFI Domain Scores. PC1—primarily represented overall sexual function; PC2—captured variation in lubrication and pain relative to the other FSFI domains. The X’s represent cluster centroids.

**Figure 5 biomedicines-13-03069-f005:**
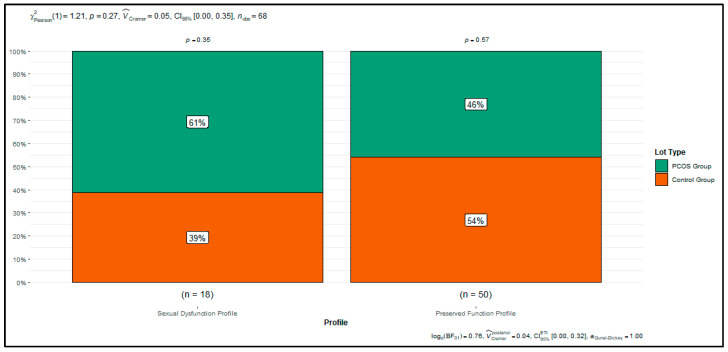
Distribution of PCOS and Control Participants Within Sexual Function Clusters.

**Table 1 biomedicines-13-03069-t001:** Sociodemographic Characteristics of the Study Population (*n* = 68). Note: All participants were sexually active within the previous 4 weeks, irrespective of relationship status.

Categorical Variable	Class	Count (Percentage)
Environment	Rural	17 (25.0%)
	Urban	51 (75.0%)
Education	General	2 (2.9%)
	High school	17 (25.0%)
	University	49 (72.1%)
Employment status	Hired	36 (52.9%)
	Not currently working	32 (47.1%)
Marital status	Married	26 (38.2%)
	Not married	42 (61.8%)
Lot	Control	34 (50.0%)
	PCOS	34 (50.0%)
FSD	Yes	35 (51.5%)
	No	33 (48.5%)

**Table 2 biomedicines-13-03069-t002:** Clinical and Psychosocial Characteristics.

Numerical Variable	Median (Q25–Q75)
Age (years)	24 (23–28)
BMI (km/m^2^)	24.55 (21.95–27.03)
MCI (days)	30 (28–89)
AC (cm)	80 (72–91)
BESAQ	1.33 (0.64–2.19)

**Table 3 biomedicines-13-03069-t003:** Female Sexual Function Index (FSFI) Scores Across Domains.

FSFI Domain	Median (Q25–Q75)
Desire	3.6 (2.9–4.8)
Arousal	4.8 (3.5–5.4)
Lubrication	4.8 (3.5–5.7)
Orgasm	4.6 (3.5–5.6)
Satisfaction	4.8 (3.2–5.2)
Pain	5.0 (3.6–6.0)
Total	26.2 (22.8–29.9)

**Table 4 biomedicines-13-03069-t004:** Cluster Stability Assessed by Bootstrapped Jaccard Index (1000 bootstrap resamples).

Cluster	Jaccard Index
1	0.90
2	0.75

**Table 5 biomedicines-13-03069-t005:** Multivariate Tests Confirming Separation Between Clusters.

Test	Statistic	Value	F	Df	*p*-Value
MANOVA	Pillai’s Trace	0.690	22.6	6, 61	<0.001
PERMANOVA	R^2^	0.288	26.7	1, 66	<0.001

**Table 6 biomedicines-13-03069-t006:** Comparison of Clinical and Psychosocial Variables Between Clusters.

Variable	Sexual Dysfunction Profile	Preserved Function Profile	*p*-Value
FSFI total	17.87 ± 5.21	28.09 ± 3.04	**<0.001**
BMI	29.13 ± 7.34	24.80 ± 4.45	**0.03**
Age	27.28 ± 5.63	25.56 ± 3.82	0.24
AC	93.89 ± 19.33	80.18 ± 11.65	**0.009**
MCI	99.11 ± 83.11	57.64 ± 55.49	0.06
BESAQ	2.48 ± 1.08	1.17 ± 0.85	**<0.001**

**Table 7 biomedicines-13-03069-t007:** Comparison of Sociodemographic Variables Between Clusters.

Variable	Class	Sexual Dysfunction Profile	Preserved Function Profile	*p*-Value
Environment	Rural	2 (12%)	15 (88%)	0.11
	Urban	16 (31%)	35 (69%)	
Education	General	1 (50%)	1 (50%)	0.69
	High school	5 (29%)	12 (71%)	
	University	12 (24%)	37 (76%)	
Employment status	Hired	10 (28%)	26 (72%)	0.80
	Not currently working	8 (25%)	24 (75%)	
Marital status	Married	6 (23%)	20 (77%)	0.62
	Not married	12 (29%)	30 (71%)	

**Table 8 biomedicines-13-03069-t008:** Logistic Regression Predicting Sexual Dysfunction Based on Cluster Membership.

Predictors	OR	CI	*p*-Value
Cluster 2	−0.66	−0.88–−0.44	<0.001
R^2^ Nagelkerke = 0.368			

**Table 9 biomedicines-13-03069-t009:** Spearman Correlations Between Sexual Function, Body Image, and Clinical Variables.

Variables		Rho	*p*-Value
FSFI total	BMI	−0.374	**0.002**
FSFI desire	Age	−0.357	**0.003**
FSFI arousal	BMI	−0.474	**<0.001**
FSFI arousal	MCI	−0.365	**0.002**
FSFI arousal	AC	−0.362	**0.002**
BESAQ	FSFI total	−0.580	**<0.001**
BESAQ	FSFI desire	−0.445	**<0.001**
BESAQ	FSFI arousal	−0.645	**<0.001**
BESAQ	FSFI orgasm	−0.437	**<0.001**
BESAQ	FSFI satisfaction	−0.432	**<0.001**

**Table 10 biomedicines-13-03069-t010:** Post Hoc Power Analysis Results.

Comparison	Effect Size	Test Type	α	Power (1-β)
FSFI Total Between Clusters	Cohen’s d = 2.74	Independent Samples *t*-test	0.05	1.00
Logistic Regression (Cluster → FSD)	Cohen’s f^2^ = 0.582	Logistic Regression	0.05	0.878

## Data Availability

The data presented in this study are available on request from the corresponding author. The data are not publicly available due to reasons concerning privacy of the subjects.

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
