# Peer review of "Multivariate Profiles of Female Sexual Function: A Cluster Analysis of FSFI Domains in Women with and Without PCOS"

_biomedicines, 2025, doi:10.3390/biomedicines13123069_

Round 1
Reviewer 1 Report
Comments and Suggestions for Authors
This is a very compelling, well-written paper that utilizes k-means cluster analysis to identify distinct profiles of sexual dysfunction among young women (n = 68, median age = 24) with or without a medical diagnosis of polycystic ovarian syndrome (PCOS). I agree with the investigators that person-centered methods have great clinical value, as they respect the heterogeneity inherent to the data and use this information to examine subpopulations which may have their own risk factors for a given health condition. The Introduction elegantly describes the state of knowledge relating PCOS to sexual dysfunction, including key elements such as excess adiposity, insulin resistance, body image dissatisfaction, and disruption of androgen-estrogen ratios. Moreover, the rationale for the study is described effectively and with a good level of detail (page 2, lines 88-94). The results of this study are presented clearly, using multiple tables, figures, and descriptive text to summarize the key findings. Overall, I have only a few minor suggestions for improvement.
(1) In Section 2.1, the text mentions that women with diabetes mellitus were excluded from the study (page 3, line 110). However, it is later stated in Section 2.2 that markers of insulin resistance were not examined in this study (page 3, line 139). Does this indicate that only women with a previous diabetes diagnosis were excluded, since glucose or insulin levels were not measured during the study period? Could any of the women have prediabetes in the study and, if so, how might that affect the interpretation of the findings (e.g., elevated blood glucose interfering with vaginal lubrication)?
(2) Under the Data Collection and Instruments section (Section 2.2), the text does not explain how the total score for the Body Exposure during Sexual Activities Questionnaire (BESAQ) is calculated. For example, how many items does this instrument contain? Is the instrument divided into multiple domains, similarly to the FSFI?
(3) Under the Statistical Analysis section (Section 2.4), please consider adding an in-text citation for the advantages of k-means clustering compared to hierarchical clustering for your particular sample size and characteristics (page 4, lines 154-155).
(4) Regarding the cluster centroids, can you please provide 1 sentence or a short phrase explaining why 68% confidence intervals are used in this situation (page 4, line 172)? Is this intended to show that the multivariate centroids are separated by at least one standard deviation unit?
(5) In Table 1 (page 5), the data show that more than 60% of women in the sample were not married. However, it is not clear from the table whether these unmarried women were dating, in long-term relationships, etc. Did all women in the study have to be in a current sexual relationship, or could they report FSFI scores from their most recent sexual relationship (e.g., during the previous 6-12 months)?
(6) For Table 2 (page 5, line 213), it may be helpful to specify the units for abdominal circumference (such as centimeters or inches). Additionally, the menstrual cycle interval unit should also be specified (e.g., days).
(7) For the Jaccard stability estimates (Table 4, page 8), please consider adding the number of bootstrap resamples to the table caption (e.g., k = 20,000). It would also be helpful to include this information in Section 2.4 (page 4, lines 163-165).
(8) For Table 8 (page 10, line 323), it may be beneficial to include the odds ratio instead of the logit regression coefficient. Alternatively, you can include both quantities in the table, or just present the odds ratio in the written text (lines 317-318).
(9) The Discussion section effectively summarizes the research findings and offers suggestions for future work. One point that might be helpful to add in this section comes from Table 6 (page 9, line 299): reading through the results, I noticed that abdominal circumference seems to more clearly differentiate between the two clusters compared to BMI. It might be worthwhile to mention how different storage sites for adipose tissue (e.g., visceral vs. subcutaneous) confer different risk profiles for both metabolic and sexual dysfunction.
Overall, excellent work on this paper! It was a pleasure to read.
Reviewer 2 Report
Comments and Suggestions for Authors
PCOS is a well studied sexual dysfunction with many different aspects. The authors tend to describe the psychological status of the women and its interaction with the syndrome. Even though the study is well presented, I consider that the results are not of major scientific value and do not contribute to a better understanding of the pathophysiology of the disease.
Reviewer 3 Report
Comments and Suggestions for Authors
This manuscript presents a person-centered cluster analysis of female sexual function in women with and without PCOS, using FSFI domain scores. The study is well-motivated, methodologically sound, and addresses an understudied aspect of PCOS with clinical relevance. The use of unsupervised clustering to transcend diagnostic categories is a strength. However, several limitations—particularly regarding sample size and generalizability—warrant careful consideration. I recommend minor revisions before acceptance.
Major Concerns
- Sample Size and Generalizability:
The total sample size (n=68) is modest for cluster analysis, especially when split into two groups. While post hoc power is high for the primary comparison, the stability of the clusters in broader populations remains uncertain. Please explain it in the article. Such as, the sample is young, educated, and predominantly urban, which may limit the generalizability of findings to other demographic or cultural contexts.
- Clustering Methodology:
The choice of k-means clustering is justified, but the gap statistic suggested k=1 as optimal, raising questions about the naturalness of the two-cluster solution. The authors’ reliance on the elbow method and clinical interpretability is reasonable but should be discussed as a limitation. The heterogeneity within the “Sexual Dysfunction” cluster (lower silhouette score) suggests that this group may not be homogeneous, which could mask important subtypes.
- Lack of Biological Markers:
The absence of hormonal (e.g., testosterone, SHBG) or metabolic (e.g., HOMA-IR) data limits the ability to explore biological mechanisms underlying the clusters, especially given the known endocrine dysregulation in PCOS.
- Causal Inference:
The cross-sectional design precludes causal interpretations. While adiposity and body image are correlated with sexual dysfunction, the direction of influence remains unclear.
Minor Concerns
- Presentation of Results:
The PCA plot (Figure 4) is informative but could be enhanced with clearer labeling and interpretation of the principal components.
Table 9 contains a typo (“IM” should be “BMI”) and could be better organized for readability.
- Discussion Section:
The authors appropriately acknowledge limitations but could more explicitly discuss the potential for unmeasured confounders (e.g., relationship quality, psychological comorbidities).
The clinical implications of the profiles could be expanded—e.g., how might these clusters inform personalized treatment approaches?
Reviewer 4 Report
Comments and Suggestions for Authors
1. The introduction section could briefly justify the use of cluster analysis over traditional group comparison methods.
2. The lack of hormonal and metabolic biomarkers (e.g., testosterone, HOMA-IR) limits the mechanistic interpretation of the findings, particularly with respect to PCOS phenotypes.
3. What does the abbreviation "IM" represent in Figure 9? Please clarify.
4. In the discussion section, the exploration of the mechanism underlying the "obesity–body image–sexual function" pathway could be strengthened by integrating relevant literature to explain why these factors specifically influence sexual arousal and lubrication. Additionally, providing specific clinical recommendations at the end of the article would enhance its clinical applicability and guidance value.
